# Diagnosing Automotive Damper Defects Using Convolutional Neural Networks and Electronic Stability Control Sensor Signals

**Thomas Zehelein** *[ID], **Thomas Hemmert-Pottmann**[ID] and **Markus Lienkamp**[ID]

Institute of Automotive Technology, Technical University of Munich, Boltzmannstr. 15,
85748 Garching, Germany; thomas.hemmert-pottmann@tum.de (T.-H.P.); Lienkamp@ftm.mw.tum.de (M.L.)
* Correspondence: thomas.zehelein@tum.de

**Abstract:** Chassis system components such as dampers have a significant impact on vehicle stability, driving safety, and driving comfort. Therefore, monitoring and diagnosing the defects of these components is necessary. Currently, this task is based on the driver's perception of component defects in series production vehicles, even though model-based approaches in the literature exist. As we observe an increased availability of data in modern vehicles and advances in the field of deep learning, this paper deals with the analysis of the performance of Convolutional Neural Networks (CNN) for the diagnosis of automotive damper defects. To ensure a broad applicability of the generated diagnosis system, only signals of a classic Electronic Stability Control (ESC) system, such as wheel speeds, longitudinal and lateral vehicle acceleration, and yaw rate, were used. A structured analysis of data pre-processing and CNN configuration parameters were investigated in terms of the defect detection result. The results show that simple Fast Fourier Transformation (FFT) pre-processing and configuration parameters resulting in small networks are sufficient for a high defect detection rate.

**Keywords:** automotive; damper; convolutional neural networks; fault detection; diagnosis; machine learning; deep learning

---

## 1. Introduction

Ensuring driving safety and driving comfort when operating vehicles requires their health state to be properly monitored. This is especially critical for chassis system components such as dampers. Currently, in addition to the driver's perception, there is only periodic human inspection for monitoring the vehicle's chassis system health state. However, this is error-prone, expensive, and implies periods of unmonitored driving between inspections. Furthermore, autonomous driving implies that the driver is not needed as a monitoring instance, either for the actual driving task or for monitoring the vehicle's health state. Therefore, an automated system for this task is necessary.

Approaches in the field of Fault Detection and Isolation (FDI) can be categorized as reliability-based, model-based, signal-based, and statistical-based FDI [1]. Existing approaches in monitoring the health of the chassis system of a vehicle are often model-based [2–4] or signal-based [5,6]. However, to the authors' knowledge, there is no such approach applied in a series production vehicle. Possible reasons are that either additional sensors that are not part of a vehicle's standard sensor set (e.g., vertical acceleration sensors) are necessary or measurements at a test-bench are required. One problem in automotive damper defect diagnosis during actual driving is robustness with regard to the vehicle's configuration, e.g., changing tire characteristics, mass variations in the vehicle, or varying road excitation. Even though driving data incorporate these different vehicle configurations and are generated while driving, the named approaches cannot benefit directly from more data. The named approaches need to get fine-tuned, which is time-consuming to match the data.

Data-driven approaches are based on measurement data that are available from a process [7]. Combining a signal-based with a data-driven approach leads to machine-learning algorithms. Robustness is therefore automatically incorporated when the supplied training data cover variations of circumstances regarding different vehicle configurations and usage scenarios. A machine-learning approach for automotive damper health monitoring using a Support Vector Machine (SVM) for classifying signal features is presented in [8]. One downside of this approach is the fact that features that can distinguish between different health states are required. Engineering representative features is therefore necessary, which is also time-consuming and requires system knowledge.

Deep learning classification architectures are able to learn features directly from the input data. The parameters of the network are adjusted with respect to minimizing a cost function that accounts for the classification result. Therefore, the overall algorithm is trained with respect to distinguishing between different states of the data. However, adding their increased complexity compared to traditional machine learning algorithms (e.g., SVM) makes sense only if these simple algorithms are lacking performance. Based on the classification results in [8,9], applying deep learning algorithms for damper defect detection should be investigated.

Convolutional Neural Networks (CNN) are stated to be able to deal with multidimensional data as well as having a good local feature extraction [10]. An overview of the different applications of CNN regarding machine health monitoring is given in [11]. CNN have emerged into a broad variety of fields, such as predictive maintenance [12,13] and medical [14,15], or mechanical diagnosis [16–23]. The latter has been dominated by model-driven approaches for decades and more recently, data-driven approaches based on feature engineering. In the past couple of years, researchers have investigated and successfully employed CNN's feature learning capabilities to specifically diagnose rotating mechanical applications such as bearings. However, to the best of the authors' knowledge, there is no application of CNN for the diagnosis of automotive suspension components such as dampers. It is therefore an open question whether CNN are equally suited for the diagnosis of automotive dampers using only Electronic Stability Control (ESC) system sensors and normal driving data. There might be similarities due to sensor signals coming from rotating wheels. However, there are big differences from industrial bearing applications to automotive applications because there is a stochastic excitation of the vehicle caused by the road profile as well as a high variability of circumstances of vehicle usage (e.g., weather conditions or parameter variations such as mass).

This paper investigates the suitability of CNN for the diagnosis of defective automotive dampers. The current state of the art is analyzed in Section 2 with regard to pre-processing methods and network architectures. Section 3 screens various pre-processing methods. Afterwards, experiments with different parameters regarding the size of the receptive field, the size of the pooling layer as well as the network depth of the CNN architecture are conducted and the resulting kernel weights of the trained networks are analyzed. Section 4 evaluates the robustness of the generated diagnosis systems with regard to variations of the vehicle setup. The paper closes with a discussion and summary.

## 2. State of The Art

This section analyzes the state of the art regarding pre-processing methods and network configurations for diagnosis applications. Hereby, many approaches deal with bearing or gearbox applications. Since CNN emerged from computer vision with two-dimensional input data such as pictures, many researchers transform their data into images. But also one-dimensional data (such as time series data or Fast Fourier Transformation (FFT) data) are used as input data to CNN.

Xia et al. [24] have classified the Case Western Reserve University (CWRU) bearing data set of [25] without any pre-processing. Acceleration sensor signals are used directly as input data to a CNN consisting of two convolutional and sub-sampling layers followed by a fully connected layer. Eren et al. [26] have also proposed no pre-processing and process the CWRU bearing dataset. A CNN consisting of three convolutional and two sub-sampling layers followed by a fully connected layer is used. Zilong and Wei [27] have also performed no pre-processing but propose a CNN architecture

that consists of multi-scale convolutional layers. Those layers incorporate the thought of "inception" modules from [28] of extracting features with convolutional kernels of different sizes in parallel. Even though it is a rather deep network architecture, the number of trainable parameters of 52,000 is still in a low range. Zhang et al. [18] have designed a CNN to operate on the noisy raw data. They claim that using Dropout, a regularization technique in the first convolutional layer helps to improve anti-noise abilities and suggest a twelve layer-deep network. Additionally, a very small batch size during training is used as well as an ensemble of CNN to further increase the classification performance.

Janssens et al. [29] have proposed using a frequency analysis as input to a CNN. Four different conditions (three different failure types and the intact state) are classified using the FFT data points of two acceleration sensors which are mounted on the bearing housing. For each condition, five bearings of the same type are used to generate the dataset for classification. Several numbers of feature maps and number of layers are tested. Lastly it is stated that a deep version of the proposed architecture is not beneficial in that use-case.

Jing et al. [17] have compared CNN-based approaches operating on raw time data, frequency analysis and time-frequency analysis as pre-processing. Different network architectures consisting of different numbers of convolutional and pooling layers are employed. The investigation was conducted based on two datasets. Both datasets consist of acceleration measurements of a gearbox housing with gears of different health states. Frequency-based data are found to work best with the proposed network. CNN architectures with fewer layers result in higher accuracies than using more layers. An increase of the input segment size results also in higher accuracies.

Gray-scale images have been employed by Wen et al. in [20] by representing each time step by a pixel with the relative signal amplitude as pixel strength. Those gray-scale images are classified using a CNN architecture that is based on the LeNet-5 architecture [30]. Their approach is tested on three different datasets, namely the CWRU bearing data set, a self-priming centrifugal pump dataset, and an axial piston hydraulic pump dataset. Other deep learning and machine learning methods such as Deep Belief Networks, Probabilistic Neural Networks, or Sparse Filter result in similar accuracies as the proposed approach.

Zhang et al. [31] have also performed pre-processing of time series vibration data by generating gray-scale images. A CNN consisting of two convolutional layers each followed by a sub-sampling layer is applied for classification. The approach is compared to using raw time signal data that are classified using a CNN and using FFT data points that are classified using a neural network.

Lu et al. [32] have proposed a nearly identical approach as in [31]. Gray-scale images are classified using a CNN with two convolutional and two sub-sampling layers. Some minor adaptions of the CNN training, such as greedy forward learning or a local connection between two layers, are proposed to increase robustness of the classification. In addition, the parameters of the convolutional and sub-sampling layers are different from [31]. To test the robustness, additional noise is added to the vibration sensor data. The proposed method achieves higher accuracy rates compared to a SVM or a shallow softmax regression classifier. However, a stacked de-noising Autoencoder results in a classification accuracy comparable to the proposed CNN approach.

Guo et al. [33] do not mention any pre-processing but transform time series data to a matrix which is in fact using a gray-scale image. The CNN for classification consists of three combinations of one convolutional and one sub-sampling layer followed by two fully connected layers.

Liao et al. [16] have compared WT and STFT as a pre-processing method for time series data. The classification is performed using a CNN that consists of two convolutional layers, each followed by a sub-sampling layer with a fully connected layer at the end of the network. Vibrational data of ten different health states of an automotive gearbox are recorded on a test bench. Using WT input data requires less training iterations compared to using STFT data.

Verstraete et al. [21] have analyzed STFT, WT, and Hilbert-Huang Transformation (HHT) as a pre-processing method for a classification using CNN. It is claimed that a STFT cannot represent

transient signals adequately while WT is effective for transient signals. The HHT is said to be suited for in-stationary signals but has numerical problems resulting in negative values under specific circumstances of the time signal. The proposed network architecture consists of two consecutive convolutional layers followed by one pooling layer. This convolution/pooling-layer combination is repeated three times and then followed by two fully connected layers. The double convolution is said to reduce the number of parameters of the network and should improve the generated features due to the additional non-linearity. The approach is tested on two bearing datasets, one of which is the CWRU bearing data set. The average classification accuracy on both datasets using the WT is slightly higher than using the STFT and the HHT has the lowest accuracy.

Zhang et al. [34] have performed a STFT on the data of the CWRU and classified them using a similar network architecture as in [21]. It consists of two consecutive convolutional layers followed by a pooling layer. In general, the overall approach in [34] is quite similar to [21], which explains the similar result.

Wavelets have also been employed by Ding and He [23] to face varying operational conditions of bearing applications. They propose a signal-to-image-approach based on Wavelet Packet Transform. This representation is used in a customized CNN that combines features from a convolutional layer and a sub-sampling layer in a special multi-scale layer. The authors claim that it enables more invariant and robust features with precise details.

The periodicity of a time signal can be visualized using a Recurrence Plot (RP) that analyzes the signal's phase space trajectory. It reflects those points at the time at which trajectories in phase space return to a previous (or close-to-previous) state. The classification performance using a RP as pre-processing method is compared to seven other time series classification algorithms based on 20 real-world datasets in [35]. The RP approach results in the lowest error rate for 10 of 20 datasets.

Another image generating pre-processing method is the calculation of Gramian Angular Fields (GAFs). They were first introduced in [36,37] for encoding time series and are used by [38] to detect defects on railway wagon wheels using a CNN architecture.

Summarizing the state of the art does not give a clear suggestion for selecting a pre-processing method or a network configuration. Various methods (such as no pre-processing, gray-scale image, STFT, WT) result in a testing accuracy above 99 % when classifying the CWRU bearing data set of [25]. Therefore, in the next section, we screen several pre-processing methods.

## 3. Conceptual Analysis Approach

Applications of CNN show promising results in the area of machine health monitoring, as shown in Section 2. However, we can find neither a favorable pre-processing method, nor do we see clear suggestions in the literature for the choice of the network architecture or its hyper-parameters, such as spatial extent, the number of kernels, or the network depth. After a description of the dataset in Section 3.1, we therefore investigate various established pre-processing methods in Section 3.2. Promising pre-processing methods are selected as input data for a network architecture investigation in Section 3.3 concerning the size for the receptive field of the convolutional layers, the size of the pooling layer as well as the network depth. To further investigate the findings regarding the network architecture, Section 3.4 investigates the CNN's feature extraction by analyzing the trained kernel weights.

### 3.1. Description of the Dataset

The analysis in this paper is based on the actual driving data of an upper class sedan vehicle of a Bavarian manufacturer with a semi-active suspension system. Defective dampers are simulated by setting constant damper currents that lead to reduced damping forces. Figure 1 shows the characteristic curve of an intact and defective damper for each axle. Even though there is a great variety of different damper defects and related consequences, simulating defective dampers by changing damper currents is a reasonable approach according to [6,39].

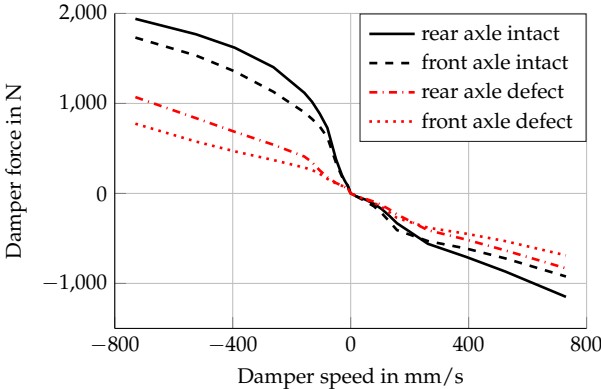

**Figure 1.** Characteristic curve of a single damper

Figure 2 visualizes the overall classification process. Although the measurement data of this paper were recorded using an upper class vehicle with semi-active suspension, it should be possible to apply the diagnosis approach to vehicles with a traditional passive chassis system. Therefore, only seven sensor signals from the vehicle's ESC system (four wheel speed signals, lateral and longitudinal accelerations as well as yaw rate) were utilized for our approach. Each sensor signal is logged with a sampling frequency of $f_s = 100\,\text{Hz}$ generating raw time signals. A sequence of 512 sequential data points is called an observation and each observation is categorized according to its damper health state. To comply with an average driving style, an observation is required to have an average longitudinal and lateral acceleration of less than $1\,\text{m/s}^2$ as well as to have an average speed above $30\,\text{km/h}$. The dataset consists of nearly 13,000 observations covering a distance of $1650\,\text{km}$ which is around $18\,\text{h}$ of driving on the German Autobahn, national and country roads as well as bad roads. The dataset is evenly distributed among the classes

- all dampers intact,
- all dampers defect,
- front left (FL) damper defect with other dampers intact and
- rear right (RR) damper defect with other dampers intact,

representing an intact suspension system, wear on all dampers due to aging, and two different single damper defects. The dataset is divided into 80 % training data and 20 % testing data. For a 5-fold cross-validation, the training data is further divided into 5 folds, whereas 4 folds are used for actual training and 1 fold is used as validation data.

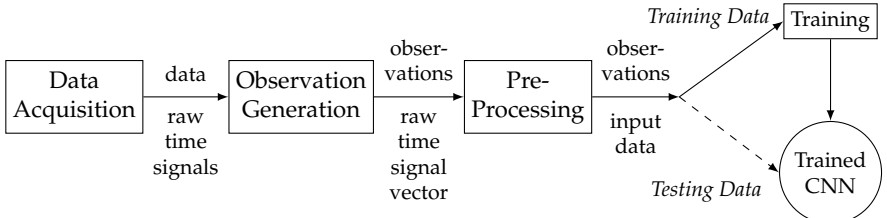

**Figure 2.** Overview of classification process

## 3.2. Analysis of Pre-Processing Methods

### 3.2.1. CNN Architecture for Pre-Processing Analysis

For the analysis of different pre-processing methods, a suitable CNN architecture needs to be defined. State of the art CNN architectures handle the first convolutional layer differently from the rest of the network. This stresses that the hyper-parameters of this layer should be chosen carefully. Furthermore, special building blocks for CNN have been proposed, e.g., Inception Modules [28,40]

or Residual Connections [41–44]. At this point, we do not know how different kernel sizes affect the results. Therefore, we use a Inception Module-like block as the first layer. This enables the network to extract features on different scales and prevents any pre-processing from suffering under unsuitable network architectures, which would lead to a decrease in performance. The Inception Module is a powerful, yet complex building block we use in our CNN and does not limit the network to a specific kernel size. The building block is depicted in Figure 3.

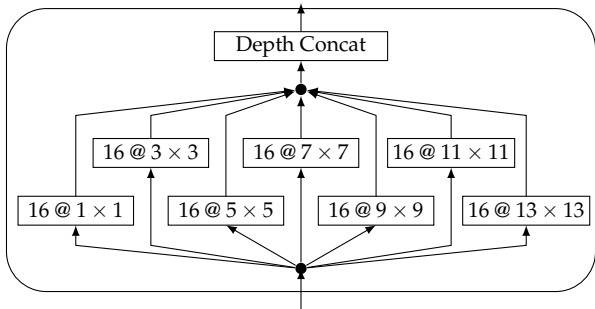

**Figure 3.** Inception-like module that is used as the first convolutional layer. The notation "16 @ 3 × 3" implies 16 filter kernels with a spatial extent of 3 × 3. All layers use *same* padding and a stride of 1.

We use seven branches with different kernel sizes and the same number of kernels for each branch. The stride is set to 1 and the padding of each convolution is chosen to be *same*, resulting in equally sized feature maps. This allows for the depth-wise concatenation of all extracted feature maps. The amount of filter kernels is chosen identically in order for there to be no one kernel size preferred over the other. Larger filters can learn lower frequencies from raw data than smaller filters, whereas small filters can be beneficial for processing peaks in the frequency spectrum data. The Inception-like module is integrated into the overall architecture given in Table 1.

**Table 1.** Architecture for the evaluation of different pre-processings

| Layer | Details |
| --- | --- |
| Input | 7 channels (4 wheel speeds, lat. & long. acceleration, yaw rate) |
| Inception-like module | See Figure 3 |
| Max-Pooling | Kernel size 2 × 2, stride 2, *valid* padding |
| Fully connected | 128 neurons |
| Dropout | Dropout rate 0.5 |
| Output | 4 neurons |

Except for the Inception-like module, the architecture of the CNN for evaluating different pre-processing methods remains simple. Max-Pooling is commonly used to establish invariance to small local changes and reduce the amount of parameters, which is why we add a single sub-sampling layer. The feature extraction stage is followed by a fully connected layer and uses dropout [45] as a simple regularization technique. The overall network architecture is shallow. Therefore, we do not make use of Batch-Normalization [46,47] or Residual Connections [41–44], which can improve convergence and significantly improve training speed in deeper networks. To prevent our network from over-fitting, we employ L2-regularization. A hyper-parameter optimization to select learning rate and L2-regularization is conducted. The cost function of this optimization is the average of the validation accuracy and the difference of training and validation accuracy to prevent over-fitting. Further details of the implementation of the neural networks are described in Appendix A.

3.2.2. Description of Pre-Processing Methods

Looking at the state of the art of CNN applications for fault detection and isolation systems of mechanical components in Section 2, many different pre-processing techniques exist. While

some publications aim at implementing end-to-end-systems, which operate on raw data, others choose simple or more complex transformations, e.g., scaling, denoising, or Fourier transformations. These transformations result in one- or two-dimensional data representations. The selected methods for investigation are chosen to represent a broad bandwidth regarding the pre-processing complexity and are explained in the following paragraphs. Figure 4 shows a data sample of the front left wheel speed $n_{\mathrm{FL}}$ that was processed with these methods.

A simple option for pre-processing is removing linear trends within the data samples. Because the driving data have been recorded at varying vehicle velocities, the magnitude of the wheel speed can be different. By subtracting a linear function, we aim at removing any bias or non-stationarity and focus primarily on transient dynamics within the signals. The linear detrend is applied before any of the other transformations we investigate.

To reduce the pre-processing effort as much as possible, simple scaling can be applied. This is recommended if the domain of multiple input channels scales differently as in this case, it speeds up the convergence of the commonly used back-propagation algorithm [48].

One-dimensional frequency-based data can be created by applying a FFT. A hanning window is applied and the one-sided spectrum is used as input data for the CNN. With a sampling frequency of 100 Hz, the maximum frequency of the FFT is 50 Hz and the input dimension is reduced from 512 data points to 256 data points per sensor signal.

Grayscale images from [20] are generated by reshaping the time signal vector to a matrix. The value of the very first data point of the time signal is indicated by the color in the top left corner and the very last data point of the time signal projected to the bottom right corner. Each row represents consecutive data points of the signal.

The STFT employs a Fourier Transformation to extract the frequency components of local sections (windows) of a signal as it changes over time. The width of the windowing function relates between frequency and time resolution. The choice of transformation parameters is decisive for the resulting size of the spectrogram. We choose a FFT segment length of 64 and an overlap of 8 samples. The resulting image is then transformed to a $32 \times 32$ pixel image. [17] suggests using a STFT rather than a combination of raw signal and FFT data. Therefore, it is employed by [16,21].

A GAF is the trigonometric sum between all points of a transformation of time series data to polar coordinates.

A phase-based two-dimensional representation, looking similar to GAFs, can be constructed by RPs [49]. The algorithm calculates a matrix norm of all data points (of a single sample) to each other and thus maps the time axis to a matrix. Hence, a signal of length $n$ becomes an image of size $n \times n$. Due to memory requirements, both the GAFs and RPs are down-sampled to a size of $32 \times 32$.

The WT as used in [23] decomposes a signal by wavelet packets [50]. The energy of the signal is calculated based on the reconstructed coefficients of the wavelet packet nodes. The resulting energy vector is then modified to a two-dimensional image according to the phase space reconstruction technique from [51].

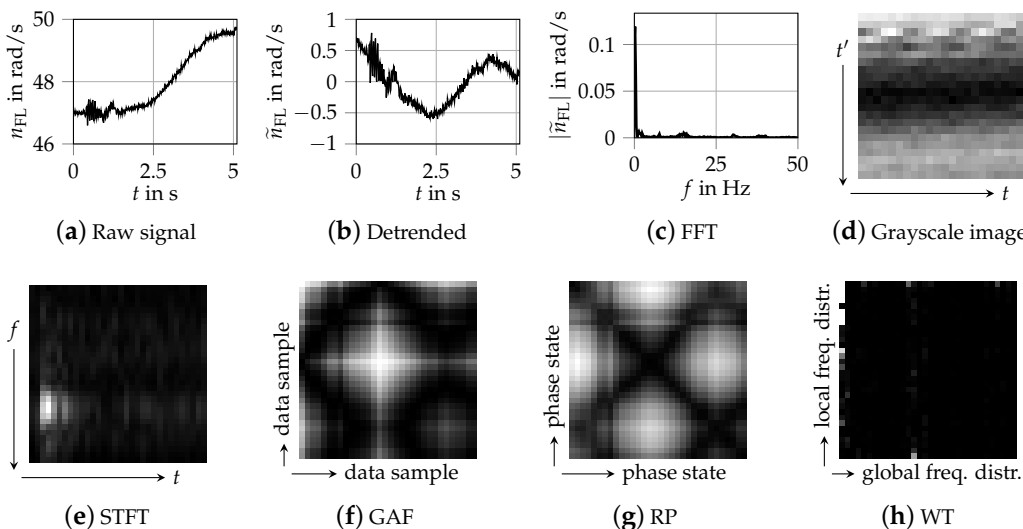

**Figure 4.** Different pre-processing representations of a data sample of the wheel speed front left $n_{FL}$.

The results for different pre-processing methods are given in Table 2. The used classification accuracy is the relation of the number of correct predictions divided by the number of observations. The statistical robustness is indicated by a mean accuracy and a standard deviation of the 5-fold cross-validation. Due to the balanced dataset consisting of four classes, randomly guessing results in 25 % classification accuracy.

**Table 2.** Test data results of a 5-fold cross-validation for different pre-processing methods

| Input Data | Pre-Processing Type | Classification Accuracy |
|---|---|---|
| 1D | None (raw signal) | $25.2 \pm 0.4\,\%$ |
| | Detrending (remove linear trend) | $79.4 \pm 1.4\,\%$ |
| | Detrending and scaling to $[-1,1]$ | $77.4 \pm 1.4\,\%$ |
| | Detrending and scaling to Gaussian Distribution | $81.2 \pm 0.5\,\%$ |
| | Detrending and apply FFT | $91.1 \pm 0.2\,\%$ |
| 2D | Detrending and apply Grayscale image | $25.0 \pm 0.0\,\%$ |
| | Detrending and apply STFT ($32 \times 32$) | $89.8 \pm 0.2\,\%$ |
| | Detrending and apply GAF ($32 \times 32$) | $32.1 \pm 1.8\,\%$ |
| | Detrending and apply RP ($32 \times 32$) | $47.1 \pm 11.7\,\%$ |
| | Detrending and apply WT ($32 \times 32$) | $72.7 \pm 1.0\,\%$ |

Operating directly on the raw data does not enable a classification due to the different vehicle speeds of the dataset. Applying a linear detrend to the time signals already improves the accuracy to nearly 80 %. Additional scaling changes this result only slightly. Applying a FFT results in the best classification accuracy. Generating gray-scale images behaves as poor as using raw data directly. The STFT results in the best classification accuracy of the two-dimensional input data versions, followed by the WT. The rather complex images generated by GAF and RP do not improve the classification. We further investigate "detrending", FFT, and STFT as a pre-processing method. Even though detrending results in a 12 percentage points (pps) lower accuracy compared to using FFT data, the performance of this simple pre-processing method using an optimized network architecture as well as the performed analyses of a CNN on time signals is of interest for the derivation of deeper knowledge of the CNN behavior.

### 3.3. Investigation of the Network Architecture

The investigation of the network architecture shall improve the classification performance and derive recommendations for the application of CNN to damper diagnosis and its hyper-parameter settings. The analysis is performed by varying the size of the receptive field $e$, the size of the max-pooling layer $p$ and the depth of the network in terms of the number of consecutive convolutional layers $d$. A first screening shows that using 16 kernels in every convolutional layer is sufficient, as some kernels tend to have very small weights and therefore, do not make any contribution to the network's output. Figure 5 shows the classification accuracy of various network configurations for detrended (first column), FFT (second column), and STFT (third column) input data using $d = 1$ convolutional layer (first row) and $d = 3$ convolutional layers (second row).

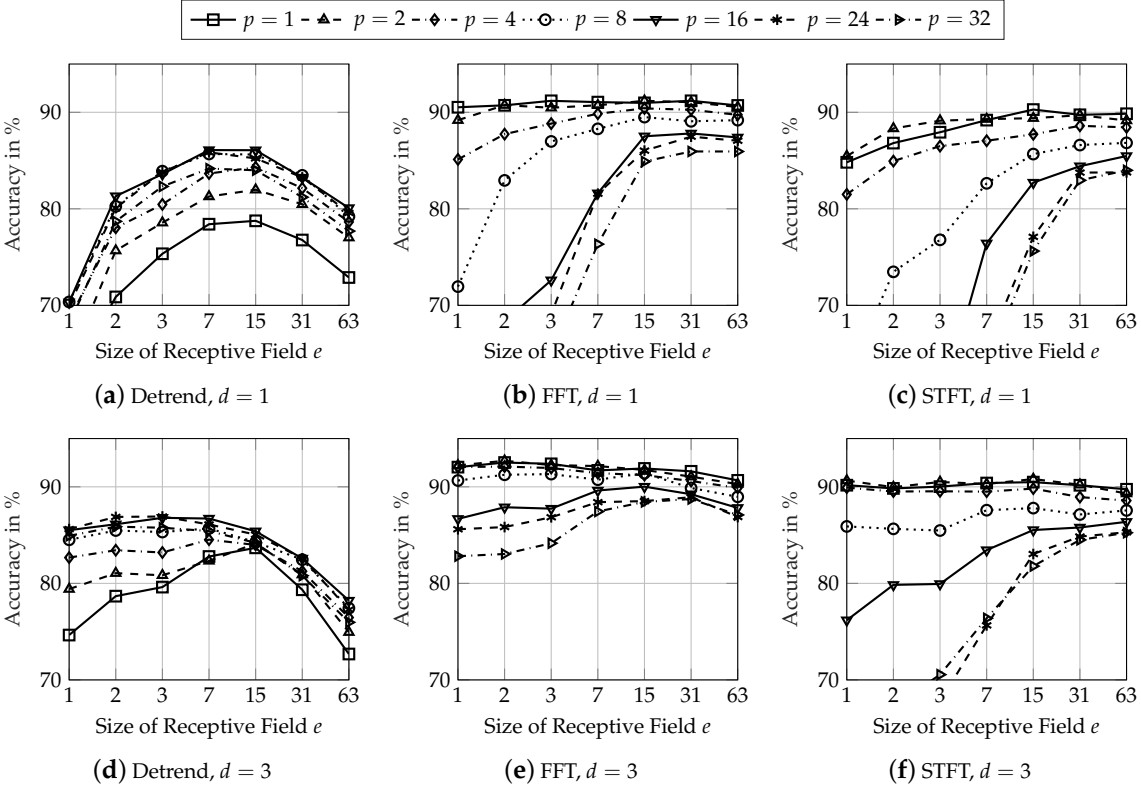

**Figure 5.** Mean test data accuracy of a 5-fold cross-validation for the investigation of the network architecture. $p$ is the size of the pooling layer (with $p = 1$ effectively resulting in no pooling), $e$ is the spatial extent of a kernel (for STFT the size of the receptive field is $e \times e$).

When using detrended input data, a greater receptive field ($e \geq 7$) is especially important. A receptive field of $e = 7$ and a data sampling frequency of 100 Hz corresponds to an oscillation frequency of approximately 14 Hz. This is approximately a chassis system's vertical eigenfrequency [52]. Model-based damper defect detection approaches also operate mainly based on this frequency [6]. Therefore, the size of the receptive field in combination with the data sampling frequency should be selected so that at least this eigenfrequency can be detected. A greater size of the pooling layer ($p \geq 8$) results in the best performance when using detrended input data. The pooling layer generates local invariance of the generated features of the preceding convolutional layer. Hereby, phase shifts between the input signal and kernel weights are compensated. This requires a size of the pooling layer of around $p = 8$ for the chassis system's vertical eigenfrequency.

When using FFT input data, the amplitude of the chassis system's vertical eigenfrequency is already included in the input data. For a small size of the pooling layer, the performance is nearly equal for different sizes of the receptive field. A pooling layer may be even disadvantageous for

frequency analysis input data if the convolutional layer compares different amplitudes at different frequencies. In fact, the smallest size of pooling layer $p = 1$ (effectively resulting in no pooling) results in the highest accuracy.

Similarly to FFT input data, the amplitudes at different frequencies are available in the STFT input data with additional time-relation information. Therefore, the behavior of using STFT input data is similar to using FFT input data. Small sizes of the pooling layer result in the best accuracy. A greater size of the receptive field increases the classification performance, especially for large sizes of the pooling layer. Even a receptive field that is larger than the actual input data ($e = 63$) results in accuracies similar to those of smaller receptive fields. However, this increases the number of trainable parameters and is therefore of no further benefit.

Additional convolutional layers improve the performance of all network configurations for every pre-processing method. The accuracies of the best performing network architectures with one convolutional layer are increased by about 1 pp. Network architectures that result in a very low accuracy with one convolutional layer have a higher increase of their classification performance with $d = 3$ convolutional layers. Due to this behavior and as additional convolutional layers generate more abstract features, the robustness regarding unknown effects in testing data might be higher for deeper network architectures.

### 3.4. Investigation of Kernel Weights

This section investigates the assumptions from Section 3.3 regarding the analyses of the input data within the neural network. Therefore, the learned kernel weights are investigated. It is assumed that supplying time-related detrended input data results in a frequency analysis, while supplying frequency analysis input data such as a FFT leads to a comparison of amplitudes at different frequencies. Figure 6 visualizes two trained kernel weights for a selected network architecture for each pre-processing method. The network architectures with depth $d = 1$ resulting in the highest accuracy was selected, which is $d = 1$, $e = 15$ and $p = 16$ for detrended input data. For FFT input data, the selected network architecture is $d = 1$, $e = 63$ and $p = 1$ as a greater size of the receptive field allows for greater insight regarding the comparison of frequency amplitudes compared to a small size of the receptive field. For STFT input data, the network architecture $d = 1$, $e = 15$ and $p = 1$ was selected as this results in the highest accuracy and the size of the receptive field still enables the interpretation of the kernel weights.

Figure 6a,d shows the weights of two kernels out of 16 for a network with detrended input data. Both kernels learned weights to analyze oscillations of the input data at different frequencies. As the shape of the weights of one kernel is similar for all input signals, each input signal is analyzed for the same frequency by one kernel. With a sampling rate of the input data of 0.01 s, Figure 6a accounts for lower frequencies (below 7 Hz), while Figure 6d accounts for higher frequencies of above 15 Hz.

Figure 6b,e shows the weights of a network with FFT input data. The kernels calculate the weighted average of the signal amplitudes at the frequencies within the receptive field. As the shape of the kernel weights consists of distinct peaks, the kernels compare amplitudes at given frequencies. The sign of the weights only matters for the following activation function but has no further physical interpretation. Sliding these peaky kernels over the FFT input data, results in a comparison of the signal amplitudes at the frequency difference indicated by the distance of the peaks. The shape of the kernel weights is similar for all signals within one kernel. Therefore, amplitudes of the same frequency differences are compared for each signal by a specific kernel. With a frequency difference between two indices of the receptive field of $\Delta f = \frac{1}{512*0.01\,\mathrm{s}} = 0.1953\,\mathrm{Hz}$, the kernel in Figure 6b analyzes for frequency differences of around $(37 - 13) * \Delta f = 4.7\,\mathrm{Hz}$ and the kernel in Figure 6e analyzes for frequency differences of around $(46 - 9) * \Delta f = 7.2\,\mathrm{Hz}$.

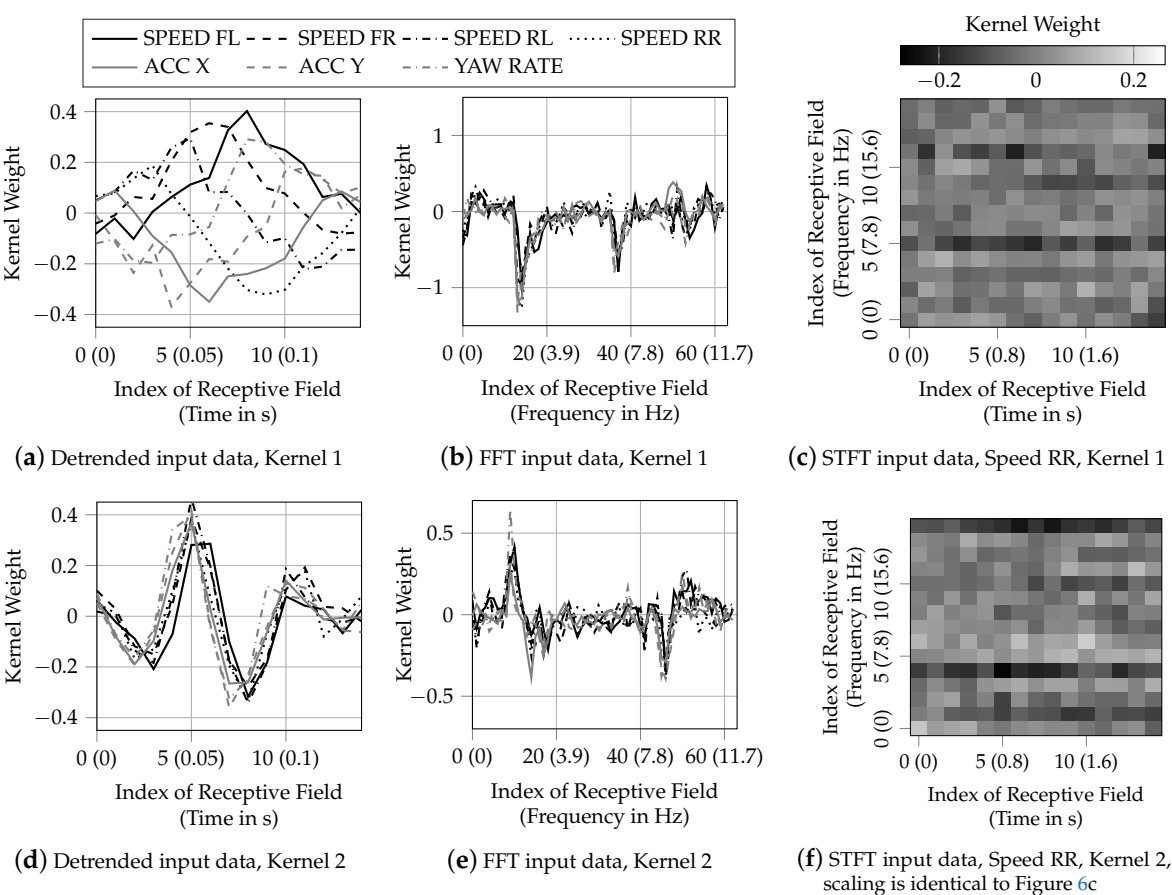

**Figure 6.** Kernel weights of the convolutional layer of different network architectures.

Two-dimensional input data results in two-dimensional kernels for each signal. Figure 6c,f visualizes the kernel weights for the rear right speed signal for a network with STFT input data. Aside from noise, noteworthy characteristics are dark horizontal lines. This demonstrates that this CNN performs analyses similar to the CNN using FFT input data by analyzing for amplitude differences at different frequencies independently from the time information on the x-axis. With the input data resolution of 32 pixels, the frequency resolution is $\Delta f = \frac{f_s}{2*32} = 1.56\,\text{Hz}$. Therefore, the kernel in Figure 6c analyzes for frequency differences of $(11 - 5) * \Delta f = 9.4\,\text{Hz}$ and the kernel in Figure 6f analyzes for frequency differences of $(14 - 4) * \Delta f = 15.6\,\text{Hz}$. While the frequency difference of 15.6 Hz has no obvious technical interpretation, 9.4 Hz are about the difference of the vehicle's body vertical eigenfrequency and the chassis's system vertical eigenfrequency. Because the time information is not used by the CNN, using a STFT is of no benefit compared to using FFT input data. It even reduces the resolution of the frequency analysis and therefore even might be disadvantageous for the classification task.

## 4. Results And Robustness

To evaluate the robustness of the trained classification systems, we created two additional different test datasets. We changed from summer to winter tires while leaving the rest of the vehicle setup unchanged. This dataset is called "tire variation". The vehicle setup of the second robustness test dataset, called "mass variation", consisted of an additional load of 200 kg in the trunk of the vehicle. Due to the package of the vehicle, this additional mass mainly affected the rear axle. Both datasets were gathered on the German Autobahn as well as national and country roads. The mass variation dataset consists of 1270 observations from 180 km. The driven roads are partly equal to the training dataset and partly different. The tire variation dataset consists of 2049 observations from 270 km,

driven on completely different roads compared to the original test dataset from Section 3. Defective dampers were simulated identically as explained in Section 3.1. The distribution of the four defective classes was nearly balanced for both datasets. In the following, the networks were trained based on the training dataset from Section 3. Therefore, effects of changed tires or additional mass are completely unknown to the trained networks. Figure 7 shows the classification accuracies of the network architectures from Figure 5 applied to the mass variation dataset and Figure 8 shows the results for the tire variation dataset.

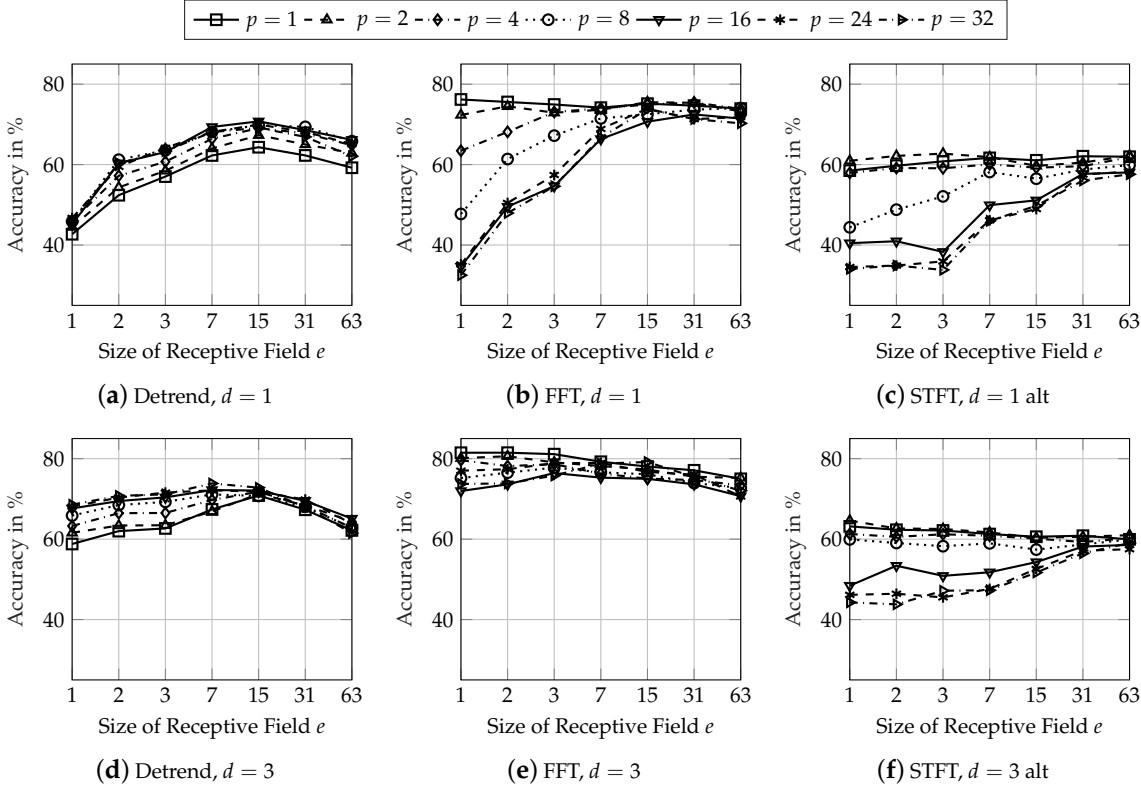

**Figure 7.** Mean accuracy of the 5-fold cross validation network architectures from Figure 5 applied to the mass variation dataset.

In general, the classification performance on both robustness datasets is lower than on the initial test dataset from Figure 5. Compared to the initial test dataset, the accuracy of the best network architecture using detrended input data is reduced by 13 pps on the mass dataset and by 23 pps on the tire dataset. When using FFT input data, the reduction of the accuracy is 11 pps on both the mass and tire dataset. Using STFT input data results in the highest reduction of classification accuracy. The accuracy on the mass dataset is 25 pps lower than on the initial test dataset and reduced by approximately 40 pps on the tire dataset.

Effects of the network architecture regarding the size of the receptive field $e$ and the size of the pooling layer $p$ are identical to the findings from Section 3.3. The accuracy is increased by about 1 pp for the best performing network architectures with an increasing depth of $d = 3$. When using FFT input data, the performance on the mass and tire datasets is increased by over 5 pps with an increase of the network depth when investigating the best network architectures from Section 3.3 ($e = 1$, $p = 1$ and $d = 1$). The classification performance using STFT input data is also increased with more convolutional layers but still remains at a low level of 50 and 60 %.

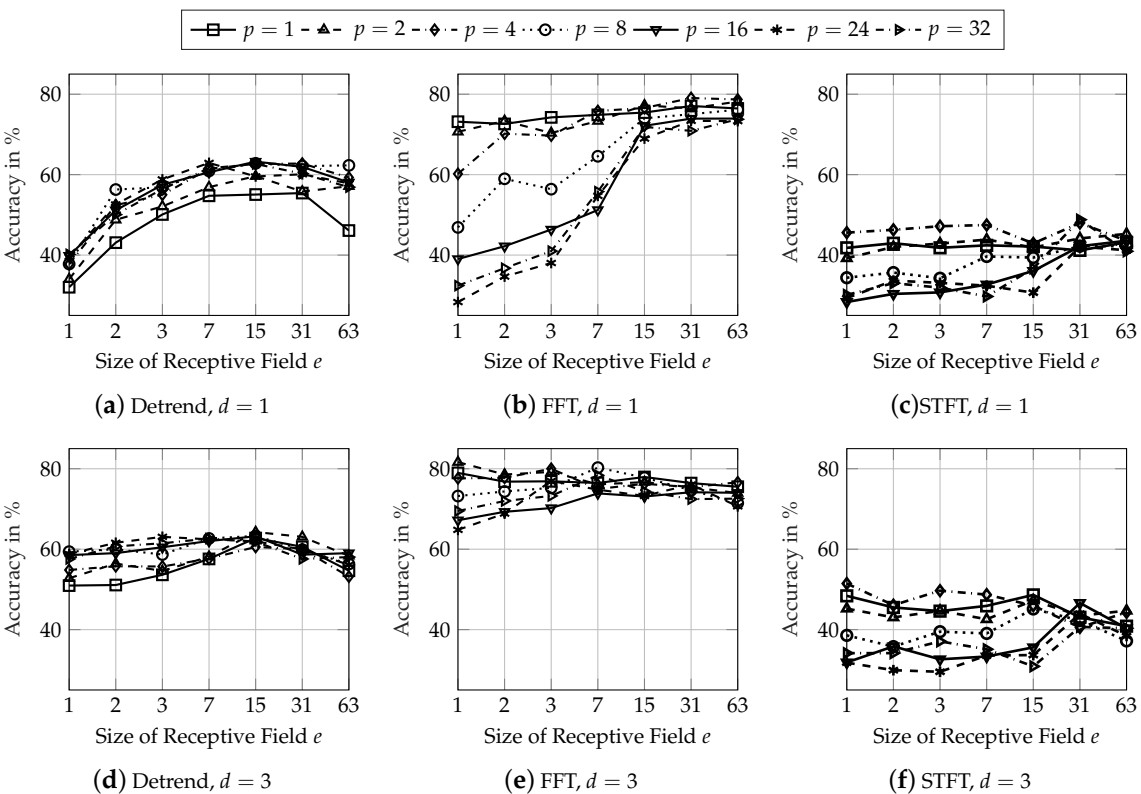

**Figure 8.** Mean accuracy of the 5-fold cross validation network architectures from Figure 5 applied to the tire variation dataset.

## 5. Discussion

The best performing network architectures using frequency analysis input data (FFT and STFT) require a small size of the pooling layer as well as a small size of the receptive field. The extreme case of $e = 1$ and $p = 1$ even results in a simple scaling of the frequency analysis of each sensor signal and averaging across the data points of all sensor signals. The actual classification is performed in the fully connected layer using those averaged frequency analysis data points. This raises the question of whether a CNN is necessary or if a Multi-Layer Perceptron (MLP) neural network is sufficient for solving the classification task. Therefore, a MLP neural network consisting of only a fully connected layer with 128 neurons was trained using the FFT input data. The resulting classification accuracy on the test dataset was 87.27 %, 72.97 % on the mass dataset and 68.37 % on the tire dataset. This demonstrates that the convolutional operation adds robustness to the performance of the neural network.

While promising results for diagnosing defective dampers using Convolutional Neural Networks are presented in this paper, limitations for a real-world implementation still exist, which will be discussed in the following paragraphs.

There are only data of one specific vehicle used in this paper's investigations. All networks were trained and tested using data of this vehicle. Recording training data for every unique vehicle configuration with different damper defects during the development phase seems challenging for an actual implementation. Therefore, portability of a trained classification system with high generalization capabilities for the diagnosis of different vehicles is desirable.

Robustness is a critical aspect for a real-world application. Even though the network architectures showed a robust behavior for tire and mass variations, further robustness analysis is necessary because there is a great variety of different circumstances during the usage of a vehicle.

Real damper defects might not occur in a switching manner, but the loss of damping forces might increase gradually over a long period of time. A classification process for the damper's health state might therefore not detect a minor defect. This can be encountered in two ways: Adding additional

classes or predicting a continuous score for the health state of each damper. However, both approaches raise the need of additional training data.

## 6. Conclusions

This paper analyzed the suitability of using Convolutional Neural Networks for the diagnosis of automotive damper defects using driving data of the longitudinal and lateral acceleration as well as yaw rate and wheel speeds. The classification performance using different pre-processing methods was analyzed. Using detrended time-signals as well as frequency analyses such as FFT or STFT showed the best results.

The analysis of the network architecture showed that the size of the receptive field and the size of the pooling layer needs to be chosen according to relevant oscillation frequencies of the input signal when using time-related detrended input data. The analysis of the trained kernel weights demonstrated that a frequency analysis is performed by the CNN for detrended time-signal input data.

Using FFT input data results in the overall best classification performance. A small size of the pooling layer performs best and the size of the receptive field can be chosen arbitrarily. The trained kernels perform a comparison of the amplitudes for several frequency differences.

Using STFT input data results in a similar classification performance as using FFT input data. The investigation of the trained kernels showed that the time information is not used by the CNN. Therefore, the STFT pre-processing does not result in any benefit. The reduced frequency resolution compared to a FFT pre-processing even decreased the robustness regarding unknown characteristics in the testing data such as additional mass or changed tires.

Table 3 shows the performance of the best network configurations of the three investigated pre-processing methods. The number of Floating Point Operations (FLOPs) for the execution of the model as well as the number of tuneable parameters are an indicator of the possibility of an implementation on the Electronic Control Unit (ECU) of a vehicle. However, since a specific value for the computing power of an automotive ECU cannot be found, the authors are not able to judge about the real-time implementation. The selected network configurations were chosen with regard to the performance of the three different datasets. The best classification accuracy results from using FFT input data with less network parameters than when using STFT input data.

The software and data of this paper are available online [53] (see the Supplementary Materials).

**Table 3.** Comparison of best performing network configurations

| Pre-Processing | Network Architecture | Tuneable Parameters | Number of Model FLOPs | Mean Accuracy in % | | |
|---|---|---|---|---|---|---|
| | | | | Test | Mass | Tire |
| Detrended Time-Signals | $e = 15, p = 16, d = 3$ | 69.444 | 485k | 85.40 | 72.08 | 63.21 |
| Detrending with a FFT | $e = 1, p = 2, d = 3$ | 264.484 | 3.67M | 92.22 | 80.14 | 81.65 |
| Detrending with a STFT | $e = 1, p = 1, d = 3$ | 2.102.564 | 14.7M | 90.17 | 63.17 | 48.40 |

**Supplementary Materials:** The software and data is available online at [53]. https://github.com/TUMFTM/Damper-Defect-Detection-Using-CNN/.

**Author Contributions:** T.Z. (corresponding author) initiated this paper. His contribution to the overall methodology of the proposed approach was essential and he performed the analyses presented in this paper. T.Z. supervised the master thesis of T.H.-P. who made essential contributions to the implementation of the proposed approaches. Both implemented this paper. M.L. made an essential contribution to the conception of the research project. He revised the paper critically for important intellectual content. M.L. gave final approval of the version to be published and agrees to all aspects of the work. As a guarantor, he accepts responsibility for the overall integrity of the paper. Conceptualization, T.Z.; methodology, T.Z. and T.H-P.; software, T.Z. and T.H-P.; validation, T.Z. and T.H-P.; formal analysis,T.Z. and T.H-P.; investigation, T.Z. and T.H-P.; resources, T.Z.; data curation, T.Z.; writing–original draft preparation, T.Z. and T.H-P.; writing–review and editing, T.Z.; visualization, T.Z. and T.H-P.; supervision, T.Z. and M.L.; project administration, T.Z.; funding acquisition, M.L. All authors have read and agreed to the published version of the manuscript.

**Funding:** The work described in this paper was supported by the basic research fund of the Institute of Automotive Technology from the Technical University of Munich. This work was supported by the German Research Foundation (DFG) and the Technical University of Munich (TUM) in the framework of the Open Access Publishing Program.

**Acknowledgments:** We would like to thank the vehicle manufacturer for supplying us with a research vehicle.

**Conflicts of Interest:** The authors declare no conflict of interest.

## Appendix A. Implementation Details

The CNN were implemented in TensorFlow 1.9 using Python 3.5. During training, we use Adam optimizer [54] with a mini-batch size of 128. The optimizer minimizes the sum of a Sparse Softmax Cross Entropy plus the sum of weight decay (L2 regularization). Training is stopped after 650 epochs or if the accuracy with validation data has not significantly improved for 50 epochs (Early Stopping). Kernels and weights are initially set following He Initialization scheme [55] and the bias terms are initially set to a small constant (0.01). We use Rectified Linear Unit (ReLU) activation functions for every layer except for the output units, which are linear neurons.

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
