# Peer review of "Diagnosing Automotive Damper Defects Using Convolutional Neural Networks and Electronic Stability Control Sensor Signals"

_jsan, doi:10.3390/jsan9010008_

Round 1

Reviewer 1 Report

In the submitted work, the authors present a method for supervised classification of the health state of a vehicle damper. For solving this task, the authors propose a Neural Network-based method that takes measurements from an Electronic Stability Control system as input and predicts the health state of the dampers. They recorded three different datasets for training and testing their proposed method. The evaluation comprises different input pre-processing methods, as well as different network architectures.

Overall, the writing style and language are appropriate. The method itself is interesting, since it follows the current trend of applying Machine Learning to different problems. However, some implementation details necessary for reproducing the results are missing, and the structures needs to be improved. Specifically:
- Sec. 2: The CWRU dataset is discussed at length, but it is actually irrelevant to the method since the CWRU focuses on bearings, whereas the work at hand focuses on dampers. Thus, this section could be shortened and Tab. 1 removed. Also, consider removing all the given accuracy numbers and focus more on a higher level of general approaches to fault detection, if there is a lack of related works on vehicle dampers.
- The evaluation and presentation of the method are tightly interwoven. It would be easier for the reader and also better for highlighting the contribution to have a distinct section with the presentation of the authors contributions/method and separate this from the evaluation part.
- Please consider properly introducing the evaluation metrics. Also consider different evaluation metrics such as False Positive Rate, False Negative Rate, etc., which are more relevant in practice than "accuracy" alone. For example, a false alarm (FPR) could trigger an irrelevant and costly inspection at a garage.
- Sec. 3.1: The authors simulate the defective dampers by adjusting the damper settings of the car. However, up to which extend does this correspond to REAL damper faults? Why are not real defective dampers used? How are the observations selected from the time series?
- Sec. 3.2: The design and training of the Neural Networks are missing important details. What is the input and the output of the networks? Which loss was used for training? Which activation functions are used? What framework is used for implementation?
- Sec. 3.2.2: The authors use timeseries data, but how is it actually represented? Does the testing use a sliding window or predefined crops? Are the crops aligned to the "events" (damper actions)? How is the temporal component represented? Did the authors consider using a Recurrent Neural Network for handling the temporal sequences?
- Sec. 3.2.2: What are the used parameters of the FFT/STFT?
- L275: The authors mention "256 datapoints per signal", but what exactly is meant by "points" and "signal"?
- Tab. 3: What exactly does "accuracy" refer to? Is it classification accuracy?
- L300ff: Consider using "%" instead of "percentage points (pps)".
- Fig. 6: Please specify the time units on the x-axis.
- L370: The authors mention that it "is highly likely that other kernels analyze for differences at the vehicle’s eigenfrequencies". Instead of being vague, please provide evidence for this, or remove the claim.
- Sec. 3.4: The analysis mentions "frequency differences", but in Fig. 6b both weights are negative, for example. I would suggest to rephrase this such that the network weights "focus" on different frequencies. Fig. 6c/f show that all weights are positive. Is this a coincidence, is it enforced during training, or are specific transformations applied for visualization?
- Fig. 8: Please consider using the same scale on the y-axis of all plots to make a comparison easier.
- Sec. 5: The authors mention problems with "minor defects" that cannot be easily detected. Why not predict a continuous score for the health instead of a discrete health state?
- Tab. 4: The results are interesting, since they show that less network parameters can be used together with FFT pre-processing in order to obtain the best results.
- What is the runtime of the network? Would it run in realtime on a car's hardware platform?
- L152: What does "double convolution" mean?
- L190/346: Instead of "chapter", use "Section" for a paper.

Reviewer 2 Report

this is a good paper. just one comment: Grayscale image has not been introduced in the text in Section 3.2.2

thanks

Round 2

Reviewer 2 Report

thanks. the authors have addressed all question.